# Follicular Fluid Amino Acid Alterations in Endometriosis: Evidence for Oxidative Stress and Metabolic Dysregulation

**DOI:** 10.3390/biomedicines13112634

**Published:** 2025-10-27

**Authors:** Csilla Kurdi, Dávid Hesszenberger, Dávid Csabai, Anikó Lajtai, Ágnes Lakatos, Rita Jakabfi-Csepregi, Krisztina Gödöny, Péter Mauchart, Ákos Várnagy, Gábor L. Kovács, Tamás Kőszegi

**Affiliations:** 1János Szentágothai Research Center, University of Pécs, Ifjúság u. 20, 7624 Pécs, Hungary; kurdi.csilla@pte.hu (C.K.); jakabfi-csepregi.rita@aok.pte.hu (R.J.-C.); kovacs.l.gabor@pte.hu (G.L.K.); 2Department of Laboratory Medicine, Medical School, University of Pécs, 7624 Pécs, Hungary; david.hesszenberger@aok.pte.hu (D.H.); csabai.david@pte.hu (D.C.); lajtai.aniko@pte.hu (A.L.); lakatos.agnes@pte.hu (Á.L.); 3National Laboratory on Human Reproduction, University of Pécs, 7624 Pécs, Hungary; godony.krisztina@pte.hu (K.G.); mauchart.peter@pte.hu (P.M.); varnagy.akos@pte.hu (Á.V.); 4Department of Obstetrics and Gynecology, Medical School, University of Pécs, 7624 Pécs, Hungary; 5MTA-PTE Human Reproduction Scientific Research Group, University of Pécs, 7624 Pécs, Hungary

**Keywords:** assisted reproduction technology, follicular fluid, amino acid profile, endometriosis, pathway analysis, oxidative stress

## Abstract

**Background/Objectives**: Endometriosis (EM) is a chronic gynecological condition associated with infertility, oxidative stress, and altered metabolic regulation. Follicular fluid (FF) reflects the microenvironment of the developing oocyte, and changes in its amino acid composition may affect reproductive outcomes. This study aimed to characterize alterations in the amino acid composition of the FF in EM and to identify potential reproductive outcomes. **Methods**: Targeted metabolomic analysis of 20 amino acids was performed on FF samples from 56 women undergoing in vitro fertilization (17 with endometriosis, 39 controls). Amino acid concentrations were quantified and compared between groups, adjusting for age and body mass index. Pathway, biomarker, and multivariate analyses were conducted to explore metabolic alterations and potential diagnostic markers. **Results**: Asparagine, histidine, and glycine concentrations were significantly higher in the EM group after adjustment for age and BMI. Pathway analysis indicated perturbations in glycine/serine metabolism, glutathione metabolism, and porphyrin metabolism, consistent with oxidative stress and mitochondrial dysfunction. Multivariate modeling demonstrated partial separation between groups, while biomarker analysis identified asparagine (AUC = 0.76), along with glycine and histidine, as potential discriminators. Additional enrichment of bile acid and methylation-related pathways suggested broader systemic metabolic changes in EM. **Conclusions**: EM is associated with distinct amino acid alterations in the FF, particularly elevated asparagine, histidine, and glycine, reflecting oxidative and mitochondrial imbalance in the follicular environment. These metabolites emerged as candidate biomarkers requiring validation for EM-related oocyte quality changes and may help individualize in vitro fertilization approaches.

## 1. Introduction

Endometriosis is a chronic inflammatory condition characterized by the presence of endometrium-like epithelial tissue outside the uterine cavity [1]. It is associated with symptoms such as pelvic pain, dysmenorrhea, infertility, dyspareunia, and dyschesia, affecting 2–10% of women in the general population and up to 40% of infertile women [2,3].

EM contributes to infertility through a combination of anatomical, hormonal, immunological, and metabolic mechanisms [4,5]. Pelvic adhesions and fibrosis may hinder oocyte release and fertilization [6], while chronic inflammation damages ovarian tissue, reduces ovarian reserve, and compromises oocyte quality [7]. Hormonal dysregulation of gonadotropins and steroid hormones can further impair ovulation and endometrial receptivity [4]. Elevated cytokines and autoantibodies in the pelvic environment disrupt gamete function and embryo transplantation [8], and increased oxidative stress exacerbates cellular damage to oocytes and embryos. Alterations in the eutopic endometrium may additionally compromise implantation even without structural abnormalities.

Follicular fluid, which fills the follicular antrum, plays a key role as a dynamic microenvironment that supports communication between germ cells and somatic cells [9]. It contains a diverse mix of hormones, proteins, antioxidants, reactive oxygen species, immune mediators, and metabolites essential for oocyte maturation [10,11]. Maintaining a balanced intra- and extraovarian environment is crucial for healthy folliculogenesis and oocyte quality [12]. Because of its accessibility during ART, FF provides a valuable window into ovarian physiology and has been increasingly explored as a source of biomarkers of oocyte quality and fertility potential [13,14].

Metabolomics, the comprehensive analysis of small-molecule metabolites, has emerged as a powerful tool in reproductive research [15]. Applied to FF, metabolomics allows assessment of oocyte metabolic status, identification of disease-related alterations, and discovery of potential biomarkers predictive of ART outcomes [16]. Importantly, recent studies have demonstrated that the metabolomic profile of FF is distinct from that of serum, highlighting FF as a localized biochemical microenvironment that more directly reflects the metabolic and functional state of the oocyte and surrounding granulosa cells [17]. Among the wide range of metabolites, amino acids play a central role in energy production, redox homeostasis, and cellular signaling during folliculogenesis. Altered amino acid metabolism has been implicated in oxidative stress and mitochondrial dysfunction (which are the hallmarks of EM-related infertility) [18].

In this study, we conducted a targeted metabolomics analysis of 20 amino acids in FF samples from IVF patients with and without EM. Using a broad range of statistical and multivariate tools available, we aimed to identify disease-specific metabolic signatures and explore their associations with clinical parameters such as age and BMI. By characterizing the altered metabolic landscape of FF in EM, our goal was to uncover potential mechanisms underlying impaired fertility and to identify candidate biomarkers or therapeutic targets that may inform clinical management and improve IVF outcomes.

## 2. Materials and Methods

### 2.1. Patient Enrollment

This study was conducted between October 2024 and March 2025 at the Department of Obstetrics and Gynecology, Medical School, University of Pécs, Hungary. All patients, or their next-of-kin, received detailed information about the study protocol, and written informed consent was obtained prior to participation. Patients under 18 years of age, as well as those who were unable to provide or withdrew consent, were excluded from the study. The study protocol was approved by the Regional Research Ethics Committee of the University of Pécs (No. 4327.316-2900/KK15/2011, approved on 26 April 2012) in accordance with the 7th revision of the Declaration of Helsinki (2013). A total of 56 patients undergoing assisted reproductive treatment (ART) were enrolled in the study, of whom 17 achieved pregnancy. Pregnancy was defined as a biochemical pregnancy, confirmed by a positive serum beta-hCG test performed 14 days post-embryo transfer and verified by the presence of a gestational sac observed via ultrasound approximately 21 days after the transfer. The EM group (*n* = 17) included patients diagnosed via ultrasound. The control group (*n* = 39) consisted of patients undergoing ART due to male infertility, tubal-factor infertility, or a combination of male and female infertility factors. Patient data were collected from medical records within the hospital’s information system.

### 2.2. Collection of the Follicular Fluid Samples

Ovarian stimulation was performed using either a long or short protocol with the GnRH agonist triptorelin or an antagonist protocol with cetrorelix. Recombinant follicle-stimulating hormone (rFSH) was administered at individualized doses ranging from 150 to 250 IU per day, based on follicular maturity. The initial dose was determined according to the patient’s BMI and age, with a maximum daily dose of 300 IU given to those identified as low responders. Depending on the patient’s age and ovarian response, stimulation was further supplemented with recombinant luteinizing hormone (rLH) or human menopausal gonadotropin (hMG). Starting from day 6 of the cycle, follicular development was monitored daily via ultrasound, and gonadotropin doses were adjusted accordingly based on follicle size. Final oocyte maturation was triggered when at least two follicles reached ≥17 mm in diameter, by administering 250 μg (6500 IU) of recombinant human chorionic gonadotropin (hCG). Oocyte retrieval was performed 36 h later using an ultrasound-guided transvaginal aspiration under routine intravenous sedation. During oocyte retrieval, follicular fluid was collected from individual follicles and pooled. Immediately after collection, the samples were centrifuged at 6700× *g* for 10 min at room temperature to remove erythrocytes, white blood cells, and granulosa cells. The supernatant was then collected and stored at −80 °C for further analysis.

### 2.3. Amino Acid Analysis

#### Sample Processing and UHPLC Analysis

Amino acids were quantified in FF following protein precipitation, fluorescent derivatization, and UHPLC with fluorescence detection (Shimadzu Nexera X2 System, Shimadzu Corporation, Kyoto, Japan) L-Norvaline was used as an internal standard. Detailed reagent specifications, derivatization steps, and chromatographic parameters (column type, mobile phases, gradients, and detector settings) are provided in the Appendix A. All measurements were performed in duplicates, and concentrations were calculated using internal standard normalized peak areas.

### 2.4. Data Analysis

Statistical analyses were conducted using SPSS for Windows (version 28.0.0.0, IBM Corp., Armonk, NY, USA). Data distribution was initially assessed for normality. Since most amino acids did not follow a normal distribution, non-parametric statistical tests were applied. Group differences were evaluated using the Mann–Whitney U-test, with a significance threshold of *p* < 0.05. In addition to *p*-values, effect sizes (r) were computed for each Mann–Whitney U test to estimate the magnitude of group differences. Amino acid concentrations were analyzed using MetaboAnalyst 5.0. (www.metaboanalyst.ca, Xia Lab, University of Alberta, Edmonton, AB, Canada; RRID: SCR_015539). Multivariate analysis was performed using Partial Least Squares Discriminant Analysis (PLS-DA) to evaluate group separation and identify key discriminative features between the EM and control groups. In addition, a heatmap with hierarchical clustering was used for unsupervised pattern recognition and to visualize relative abundance differences across samples. Biomarker analysis was conducted by generating Receiver Operating Characteristic (ROC) curves, where metabolites with an Area Under the Curve (AUC) > 0.7 were considered to have acceptable discriminatory power between groups. For functional interpretation, Metabolic Pathway Analysis was carried out using pathway enrichment and topology analysis. Pathways with a topological impact score greater than 0.10 were considered biologically relevant. To further explore biological significance, Metabolite Set Enrichment Analysis (MSEA) was applied to identify enriched metabolite sets based on known biological functions and disease associations. Relevant pathways associated with EM were visualized using metabolic network diagrams.

## 3. Results

### 3.1. Patients’ Demographics and Clinical Characteristics

Table 1 presents the demographic and clinical characteristics of the patients included in this study.

### 3.2. Amino Acid Analysis of the FF Samples

A total of 20 amino acids, the fundamental building blocks of proteins, were measured in all FF samples. As shown in Table 2, glutamine, alanine, and glycine were the most abundant amino acids. This observation is consistent with previous studies [18], which highlight the physiological importance of these amino acids in oocyte development [19,20].

### 3.3. Comparison of Amino Acid Profiles Between EM and Control Groups

The results of the statistical comparison of the two groups are shown as *p*-values and are presented in Table 3. The patients were separated into two groups based on EM disease (EM, *n*= 17) and patients without this condition (CG, *n*= 39). Three amino acids were found to be significantly altered in the EM group. These were asparagine (*p* = 0.002, r = 0.411), histidine (*p* = 0.049, r = 0.263) and glycine (*p* = 0.025, r = 0.299). The concentration of all three amino acids was higher in the EM group, corresponding to small-to-moderate effect sizes.

### 3.4. Comparison of Amino Acid Concentration Based on Body Mass Index (BMI)

The results of this comparison are presented in Table 3. Patients were categorized into two groups according to their BMI, which was calculated as weight in kilograms divided by height in meters squared (BMI = kg/m^2^). Those with a BMI between 18.5 and 24.9 were classified as the normal group (*n* = 35), while patients with a BMI above 25 were placed in the overweight group (*n* = 19). In the EM group, only 3 patients had higher BMI, and 14 patients were in the normal BMI group. This result corresponds to previous studies where it was concluded that patients with EM tend to have lower BMI [21]. When comparing the normal and high BMI groups, among the amino acids measured, only asparagine (*p* = 0.018, r = −0.317) showed a significant difference. The asparagine concentration was higher in the normal group, and the negative effect size indicates a moderate difference between the groups.

### 3.5. Comparison of Amino Acid Concentration Based on the Age of the Patients

For this analysis (Table 3), the patients were divided into two age groups: the younger group (aged 34 and below, *n* = 30) and the older group (aged 35 and above, *n* = 26). One amino acid, glycine (*p* = 0.033, r = 0.281), was significantly altered, and the concentration of this amino acid was higher in the older group. The small-to-moderate effect size indicates an age-related variation in glycine levels.

### 3.6. Heatmap and PLS-DA Analysis

Heatmaps provide a detailed view of individual metabolite patterns in complex omics data, such as the amino acid profile in FF. Heatmaps reveal relative abundance differences across samples. In this study, a heatmap of scaled concentrations of 20 amino acids in FF was used to compare EM and control IVF patients. Rows represent amino acids; columns represent individual samples and color gradients (blue to red) indicating low to high concentrations. Hierarchical clustering applied to both samples and metabolites helps uncover similarity patterns, suggesting potential biological subgroups or co-regulated metabolic pathways. Notably, altered levels of glycine, histidine, and glutamate suggest EM-related metabolic changes (Figure 1).

To enhance group separation and identify discriminative metabolic features between patients with EM and controls, Partial Least Squares Discriminant Analysis (PLS-DA) was applied to the amino acid concentration data obtained from FF. The PLS-DA score plot displays the projection of individual samples along the first two latent variables: Component 1 accounts for 45.6% of the variance related to group separation, and Component 2 explains 16.1% of the remaining discriminative variance. Each point represents an individual sample, colored by group (pink = control, green = EM), and shaded ellipses depict 95% confidence intervals for each class. The two groups show partial separation, which indicates differences in amino acid profiles that are relevant to group classification. The overlap of confidence intervals suggests metabolic heterogeneity with both groups, while the group displacement along Component 1 supports the presence of systematic metabolic differences associated with EM. Although partial separation was observed, the overlapping confidence intervals indicate that classification performance is modest, and the results should be interpreted as exploratory. This analysis suggests that specific amino acids contribute to the discrimination between the two groups. In terms of biological significance, the PLS-DA findings support the hypothesis that metabolic alterations in the follicular environment are associated with EM (Figure 2).

### 3.7. Metabolite Set Enrichment Analysis (MSEA)

The dot plot presents the results of MSEA performed on amino acid concentration data obtained from FF samples (Figure 3). The analysis shows the top 25 metabolic pathways enriched in the dataset, providing information on biological processes potentially altered in the follicular environment of patients with EM compared to controls. The x-axis represents the statistical significance of each pathway, expressed as the negative logarithm of the *p*-value (−log_10_(*p*-value)). Pathways farther to the right are more statistically significant. The y-axis lists the names of the metabolic pathways. Each dot represents one enriched pathway. The size of the dot corresponds to the enrichment ratio, which reflects the degree of overrepresentation of metabolites from the experimental dataset within a given pathway. A higher enrichment ratio indicates a stronger overrepresentation. The color gradient of the dots—from yellow to red—indicates the raw *p*-value, with deeper red representing more significant pathways. The most significantly enriched pathways include porphyrin metabolism, bile acid biosynthesis, methionine and methylhistidine metabolism, carnitine synthesis, alanine metabolism, glutathione metabolism, and glycine and serine metabolism.

### 3.8. Biomarker Analysis

To identify potential biomarkers that distinguish FF profiles between EM and control groups, a biomarker analysis was performed using ROC-based metrics. Among the 20 amino acids examined, asparagine (AUC = 0.76, *p* = 0.016), glycine (AUC = 0.69, *p* = 0.038), and histidine (AUC = 0.67, *p* = 0.042) emerged as the most promising discriminating features, as their differences were statistically significant and the AUC values were above 0.65. These findings suggest that these metabolites may serve as potential biomarkers of altered follicular microenvironments in EM. In Figure 4, the ROC curve of asparagine is demonstrated together with the results of Mann–Whitney probes. This amino acid demonstrated moderate discriminative ability, indicating its potential as a biomarker candidate within a broader metabolic context. However, its clinical applicability remains limited when considered as a single marker. Validation in larger cohorts and integration with additional metabolites or relevant clinical parameters will be essential to confirm its diagnostic or prognostic value.

### 3.9. Pathway Analysis

Pathway enrichment analysis demonstrated several metabolic pathways that were altered between the EM and control groups (Figure 5). Among these, glycine, serine, and threonine metabolism emerged as the most significantly affected pathway, showing both high statistical significance (*p* = 0.047) and substantial pathway impact (PI = 0.47). The one-carbon pool by folate and lipoic acid metabolism also showed high significance, but a moderate impact. Other notably enriched pathways were porphyrin metabolism, glutathione metabolism, beta-alanine metabolism, sphingolipid metabolism, and cysteine and methionine metabolism. Pathways with high impact but lower significance were phenylalanine, tyrosine, and tryptophan biosynthesis, as well as alanine, aspartate, and glutamate metabolism.

## 4. Discussion

This study identified distinct metabolic alterations in the FF of ART patients with EM compared to controls, emphasizing specific amino acids that reflect the disrupted follicular environment associated with the condition. Among the 20 amino acids examined, asparagine, histidine, and glycine exhibited statistically significant differences between the EM and control groups. While these metabolic differences were largely independent of age and BMI, certain associations—such as BMI with asparagine and valine, and age with glycine—suggest that individual patient characteristics may influence specific aspects of the follicular metabolome.

In FF samples, the concentration of asparagine was significantly altered. Although its role in FF has not been extensively described, increased asparagine has been reported in eutopic endometrial tissue from EM patients, likely to reflect enhanced protein synthesis and altered energy demands of proliferative endometrial cells [22]. Interestingly, asparagine concentration was found to be lower in FF samples from patients with higher BMI. Obesity is associated with systemic metabolic changes that can affect the composition of FF, which plays a crucial role in oocyte development. A study demonstrated that in individuals with high body fat, asparagine levels were reduced compared to those with normal body fat [23]. Contrarily, previous reports have shown elevated asparagine concentrations in the serum of obese patients, highlighting a possible discordance between systemic and local (follicular) amino acid levels [24].

Histidine was also found at significantly higher concentrations in the FF of the EM group. A previous untargeted study of FF samples from EM patients reported aromatic amino acids and related metabolites were upregulated, suggesting increased biosynthesis in EM, possibly in response to chronic inflammation and oxidative stress characteristic of this condition [25]. Furthermore, a systematic review found that while histidine levels did not significantly differ in plasma, they were consistently elevated in FF from women with EM. This suggests a localized, rather than systemic alteration in amino acid metabolism within the ovarian microenvironment [26].

Glycine levels were also significantly elevated in the EM group. In another metabolomics study, glycine was reported at a higher concentration in the FF of EM patients. Glycine is a key component in the formation of the antioxidant glutathione, and in increased oxidative stress, glycine may promote its utilization in GSH synthesis in EM. Elevated glycine may represent a compensatory response to oxidative stress, as supported by studies showing that glycine supplementation reduces ROS and improves mitochondrial function in oocytes [12,20]. Age-related increases in glycine may similarly reflect a metabolic adaptation to declining antioxidant capacity [27].

When amino acid profiles were compared based on BMI, valine concentrations were higher in patients with higher BMI, and this is consistent with reports showing increased branched-chain amino acids (BCAAs) in FF from women with higher BMI. Elevated BCAA concentrations in FF have been linked to impaired oocyte quality, with reduced pregnancy rates and a higher risk of miscarriage in women with obesity or insulin resistance [28].

The metabolic landscape of FF in patients with EM is profoundly altered, reflecting the complex pathophysiology of the disease. Our metabolite set enrichment analysis MSEA and pathway analysis revealed significant disruptions in several key metabolic pathways, including porphyrin metabolism, primary bile acid biosynthesis, methionine and methyl histidine metabolism, carnitine synthesis, glutathione metabolism, and glycine, serine, and threonine metabolism pathways. Many of these pathways converge on oxidative stress regulation, mitochondrial energy metabolism, and methylation processes, all of which are central to the pathophysiology of EM.

EM lesions are characterized by iron overload from retrograde menstruation, leading to excessive heme degradation and local ROS generation [29,30,31,32]. These findings align with our observation of altered heme-related metabolism and suggest that systemic oxidative imbalance extends into the follicular environment.

Our data also indicated alterations in primary bile acid biosynthesis, which further highlights the link between metabolic and hormonal dysregulation in EM. Bile acids influence estrogen signaling and immune regulation, and recent evidence suggests that gut microbiota shifts in EM may alter bile acid composition and signaling [33,34]. A recent integrative study [35] also demonstrated disrupted bile acid metabolism and its interaction with estrogen signaling and immune pathways in EM, reinforcing the importance of this metabolic–endocrine interface in disease progression. While their connections remain partly speculative, they may represent an indirect mechanism linking systemic metabolic disturbances with ovarian dysfunction.

Pathways related to methionine and methyl histidine metabolism were also significantly altered. These changes may reflect disrupted methylation capacity and altered epigenetic regulation, as overexpression of nicotinamide N-methyltransferase and other methyltransferases has been reported in EM tissues [24,36]. The enrichment of carnitine synthesis and lipoic acid metabolism points to impaired mitochondrial β-oxidation and energy production, consistent with the recognized mitochondrial dysfunction in EM [37].

Alterations in glutathione metabolism and glycine–serine–threonine metabolism suggest that EM FF is characterized by a redox imbalance and disrupted one-carbon metabolism. Glycine and serine are central to glutathione synthesis and NADPH production, and their dysregulation weakens antioxidant defense and promotes a pro-oxidant, inflammatory follicular milieu [38,39,40,41,42,43]. Serine and glycine are critical for anti-inflammatory and immunoregulatory pathways, glycine depletion impairs immune homeostasis, and both amino acids are required for rapid T-cell expansion and proper cytokine responses. Immune dysregulation associated with serine/glycine deficiency may thus contribute to persistent inflammation in EM and to age-related immune dysfunction [44].

Several methodological considerations should be acknowledged when interpreting the results of this study. Ovarian stimulation was performed using different protocols (long and short protocols) with starting doses determined individually by the treating physicians based on clinical judgment. Standardized dosing tools, such as nomograms (e.g., La Marca) or AI-based systems, were not applied. Although these methods may help minimize inter-patient variability and optimize stimulation outcomes, their implementation was beyond the scope of this work. Future studies could incorporate such approaches to further refine stimulation protocols and evaluate their potential impact on FF metabolomics and oocyte quality.

Another limitation concerns the pooling of FF samples. Fluids collected from multiple follicles were combined for each patient to ensure sufficient sample volume for metabolic analysis. While this approach enables comprehensive biochemical assessment, it may mask follicle-specific differences, as the metabolic composition of fluid from follicles containing oocytes can differ from that of empty follicles. Consequently, subtle variations related to oocyte presence or developmental potential could not be evaluated. Future research focusing on single-follicle analysis may provide deeper information into the relationship between local metabolism and oocyte competence.

The relatively modest sample size (17 EM and 39 controls) limits the statistical power of this study and may contribute to the partial separation observed in multivariate analyses. Therefore, the findings should be interpreted as exploratory and confirmed in larger cohorts.

Finally, although the identification of metabolic pathways associated with positive ART outcomes in EM would be of interest, only six of the seventeen EM patients in our cohort achieved pregnancy. This limited number precluded reliable subgroup or pathway-level analyses. Larger cohorts will be needed to validate our findings and to explore potential metabolic predictors of successful ART outcomes in this patient population.

Our findings support the concept that EM is associated with a reprogrammed FF metabolomic profile characterized by enhanced oxidative stress, iron overload, disrupted hormonal signaling, and impaired energy and methylation metabolism. These metabolic perturbations may contribute to the subfertile phenotype observed in EM and offer potential targets for therapeutic intervention or biomarker development. Integrating metabolomic findings with transcriptomic and proteomic data in future studies will be essential to further elucidate the mechanistic underpinnings of these altered pathways and their relevance to reproductive outcomes.

## 5. Conclusions

Our findings demonstrated that endometriosis is associated with significant metabolic alterations in the amino acid profiles in the follicular fluid of IVF patients. Elevated levels of asparagine, histidine, and glycine, along with disruptions in pathways related to oxidative stress, mitochondrial function, and hormonal regulation, suggest a compromised follicular environment that may negatively influence oocyte quality and reproductive potential. These results enhance our understanding of the metabolic underpinnings of endometriosis-related infertility and highlight potential biomarkers and therapeutic targets to support individualized treatment strategies and improve IVF outcomes in this patient population.

## Figures and Tables

**Figure 1 biomedicines-13-02634-f001:**
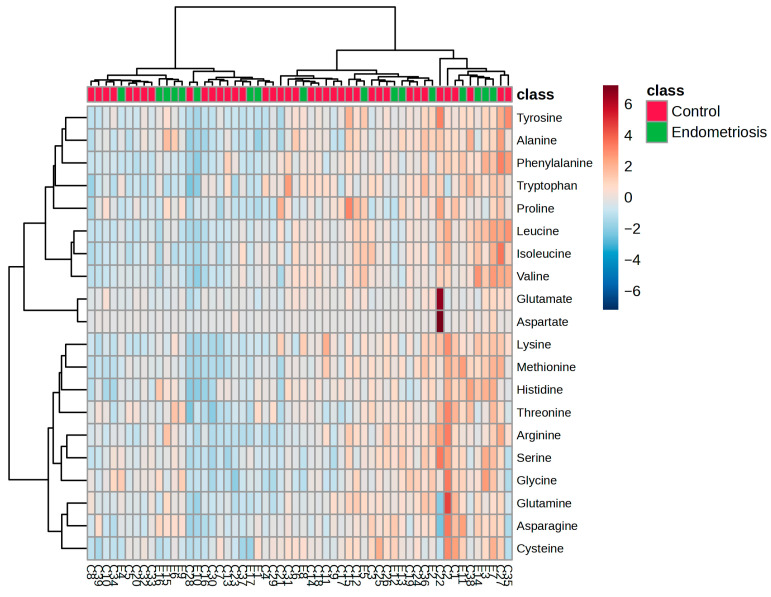
Heatmap visualization of the 20 main amino acids in FF from CG and EM patients. Each column represents an individual patient (EM, *n* = 17; CG, *n* = 39) and each row represents a specific amino acid. Hierarchical clustering was applied to both rows (amino acids) and columns (samples) to highlight similarity patterns. Colors indicate relative amino acid abundance (red = higher, blue = lower).

**Figure 2 biomedicines-13-02634-f002:**
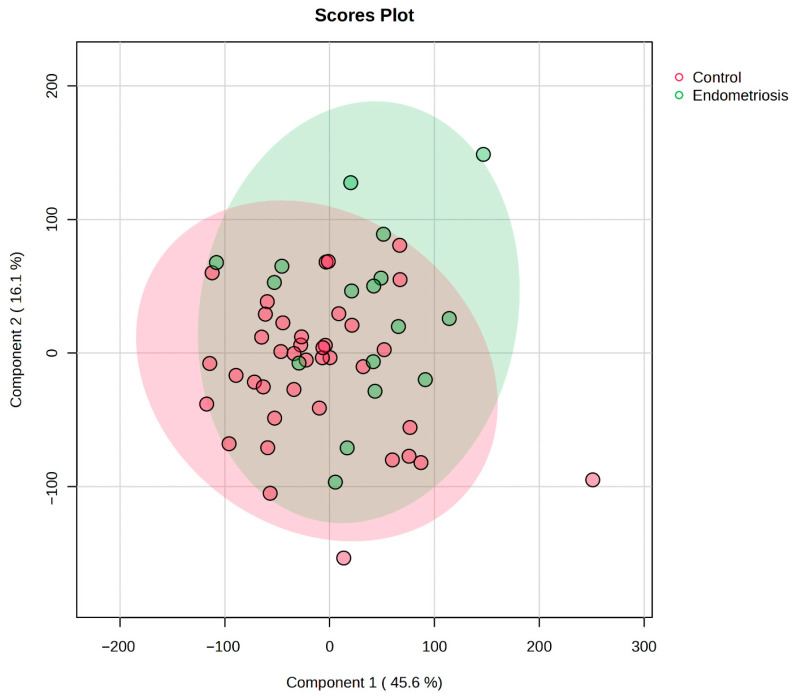
PLS-DA score plot based on FF amino acid profiles showing the difference between EM and control groups. Component 1 explains 45.6% and Component 2 16.1% of the variance. Shaded ellipses depict 95% confidence intervals. Partial separation along Component 1 indicates systematic metabolic differences associated with EM, while overlapping confidence intervals suggest metabolic heterogeneity within both groups. The background color indicates group membership: green represents control patients, while red represents the EM group.

**Figure 3 biomedicines-13-02634-f003:**
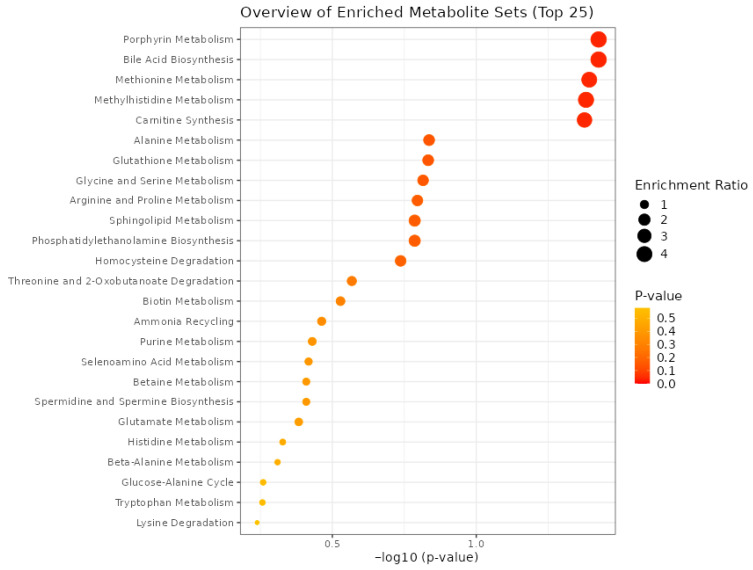
Dot plot illustrates the top 25 significantly enriched metabolic pathways identified by MSEA based on FF amino acid profiles of EM and control patients. The size of the dots corresponds to the enrichment ratio, indicating the degree of overrepresentation of metabolites from the dataset, and the higher enrichment ratios reflect stronger overrepresentation. The color gradient (from yellow to red) indicates the raw *p*-value, with deeper red representing more statistically significant pathways.

**Figure 4 biomedicines-13-02634-f004:**
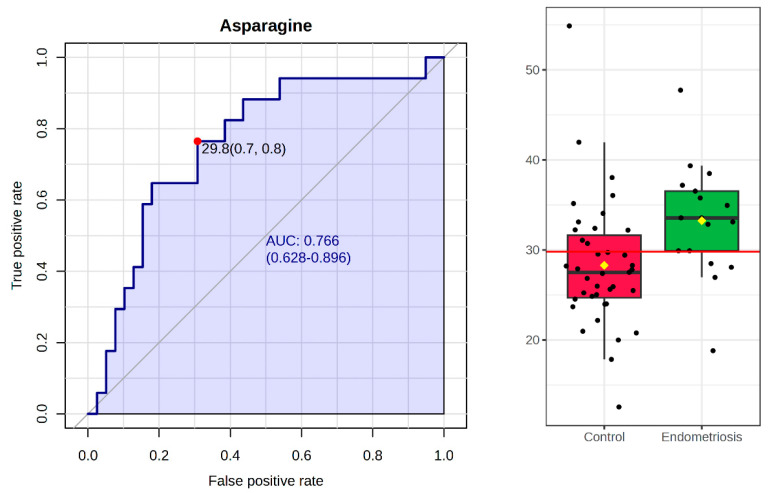
ROC curve and boxplot for asparagine in the FF of EM and control patients. Biomarker analysis of 20 amino acids identified asparagine (AUC = 0.76, *p* = 0.016) as the most promising discriminating feature. The ROC curve for asparagine demonstrates moderate discriminative ability, while the accompanying boxplot shows its distribution between the two groups.

**Figure 5 biomedicines-13-02634-f005:**
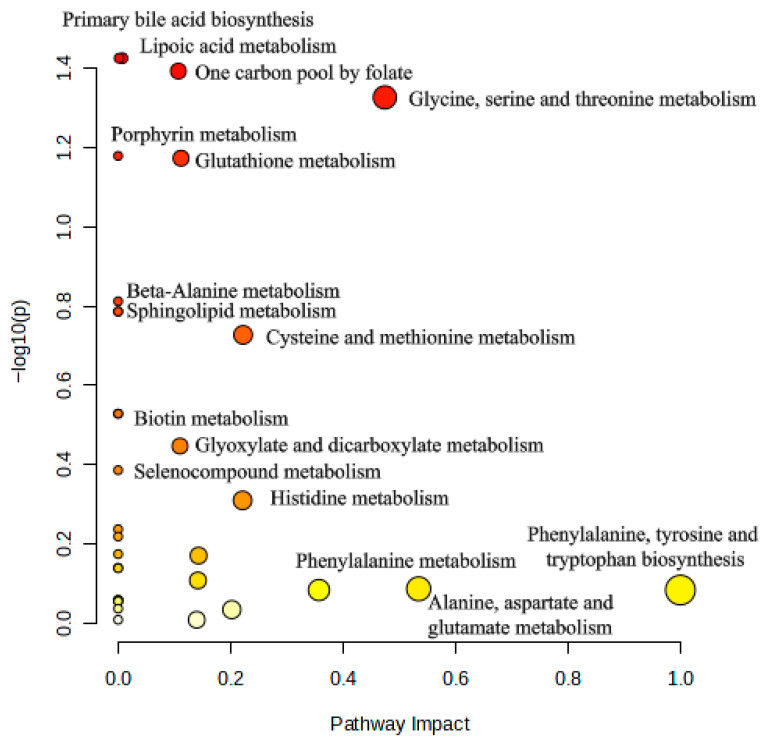
Pathway analysis of amino acid profiles in follicular fluid from EM and control patients. Glycine, serine, and threonine metabolism was the most significantly altered pathway (*p* = 0.047, PI = 0.47). Other notable pathways with high significance or impact included the one-carbon pool by folate, lipoic acid, porphyrin, glutathione, beta-alanine, sphingolipid, and cysteine and methionine metabolism. In the figure, the color intensity (from yellow to red) and the size of the dots indicate pathway significance, with deeper red and larger dots representing more statistically significant pathways.

**Table 1 biomedicines-13-02634-t001:** Clinical characteristics of the patients in the EM and control groups.

	Endometriosis [Mean ± SD] (*n* = 17)	Control [Mean ± SD] (*n* = 39)	Significance (*p*-Value)
Age	35.14 ± 4.34	33.26 ± 4.18	0.234
BMI	22.87 ± 5.71	25.29 ± 5.66	0.025
Number of oocytes retrieved	8.13 ± 4.49	10.49 ± 6.66	1.000
Number of fertilized oocytes	3.18 ± 2.00	3.89 ± 3.91	0.685
Number of IVF cycles	1.94 ± 0.93	1.86 ± 1.38	0.338
Baseline estradiol	2954.44 ± 749.75	2426.51 ± 912.06	0.04
FSH dose during stimulation	1083.94 ± 1430.84	1417.31 ± 1382.14	0.118
Cause of infertility			
Male factor	-	26 (66.66%)	
Female factor	15 (88.23%)	8 (20.51%)	
Combined male–female	2 (11.76%)	5 (12.82%)	

**Table 2 biomedicines-13-02634-t002:** Mean concentrations (±standard deviation) of 20 amino acids in FF samples. The statistically significant differences (*p* < 0.05) between the two groups are indicated with an asterisk.

	Endometriosis [Mean ± SD] µmol/L	Control [Mean ± SD] µmol/L
Aspartate	7.01 ± 4.28	8.45 ± 16.49
Glutamate	63.35 ± 20.85	69.67 ± 47.92
Asparagine *	33.25 ± 6.28	28.29 ± 7.12
Serine	57.31 ± 15.84	50.09 ± 18.28
Glutamine	350.41 ± 72.65	346.30 ± 101.53
Histidine *	58.95 ± 13.26	51.71 ± 11.33
Glycine *	178.58 ± 51.28	147.43 ± 49.84
Threonine	109.51 ± 23.00	100.46 ± 29.82
Arginine	34.85 ± 11.00	33.16 ± 13.65
Alanine	240.32 ± 56.45	226.05 ± 60.46
Tyrosine	30.82 ± 8.45	30.75 ± 11.26
Cysteine	20.09 ± 6.56	20.40 ± 6.66
Valine	134.37 ± 53.51	128.41 ± 34.81
Methionine	16.81 ± 4.39	15.75 ± 4.10
Tryptophan	47.65 ± 5.92	48.44 ± 6.70
Phenylalanine	38.52 ± 9.65	40.04 ± 9.21
Isoleucine	27.22 ± 9.19	29.42 ± 8.82
Leucine	54.74 ± 20.94	54.74 ± 18.48
Lysine	89.35 ± 21.57	81.89 ± 25.42
Proline	143.05 ± 42.37	150.10 ± 54.00

**Table 3 biomedicines-13-02634-t003:** Results of the comparison across various parameters. Each number represents the corresponding *p*-value. Statistically significant values (*p* < 0.05) are marked in bold.

Significance (*p*) Values				
	EM/CG	BMI	Age	Outcome
Aspartate	0.515	0.177	0.967	0.314
Glutamate	0.599	0.069	0.755	0.142
Asparagine	**0.002**	**0.018**	0.651	0.212
Serine	0.06	0.921	1	0.755
Glutamine	0.493	0.208	0.384	0.755
Histidine	**0.049**	0.431	0.332	0.076
Glycine	**0.025**	0.42	**0.033**	0.350
Threonine	0.173	0.48	0.56	0.612
Arginine	0.368	0.651	0.48	0.810
Alanine	0.25	0.273	0.393	0.838
Tyrosine	0.493	0.135	0.663	0.482
Cysteine	0.908	0.474	0.954	0.702
Valine	0.922	**0.042**	0.576	0.386
Methionine	0.465	0.758	0.712	0.402
Tryptophane	0.782	0.556	0.576	0.281
Phenylalanine	0.605	0.436	0.495	0.412
Isoleucine	0.222	0.13	0.332	0.515
Leucine	0.88	0.265	0.657	0.769
Lysine	0.151	0.863	0.818	0.587
Proline	0.838	0.25	0.44	0.551

## Data Availability

The original contributions presented in the study are included in the article/Appendix A, further inquiries can be directed to the corresponding author.

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
