# Peer review of "Follicular Fluid Amino Acid Alterations in Endometriosis: Evidence for Oxidative Stress and Metabolic Dysregulation"

_biomedicines, 2025, doi:10.3390/biomedicines13112634_

Round 1

Reviewer 1 Report

Comments and Suggestions for Authors

# 120  The use of two different stimulation protocols (long protocol vs. short protocol) could alter oocyte metabolism. For this reason, it would be appropriate to report the results of follicular fluid metabolomics separately in the group of women treated with the long protocol compared to those treated with the short protocol.

# 122 It is recommended to use standardized starting dose calculation methods such as the La Marca nomogram or more sophisticated AI calculation systems.

# 138 It is necessary to specify whether the follicular fluid is analyzed for each individual follicle or whether the follicolar fluids of all patients are studied together. There may be differences between FF that contain oocytes and FF that do not contain oocytes.                                                                                                                                                                                                                                   It  would be interesting to understand whether in the group of endometriosis patients with a positive pregnancy test there is a metabolic pathway predictive of a positive ART outcome compared to endometriosis patients who had a negative pregnancy test after IVF/ICSI.

Reviewer 2 Report

Comments and Suggestions for Authors

Attached

Reviewer 3 Report

Comments and Suggestions for Authors

The manuscript by Kurdi et al. examines metabolic alterations in follicular fluid (FF) associated with endometriosis (EM). By analyzing amino acid profiles and linking them to oxidative stress, mitochondrial dysfunction, and potential biomarkers, the study contributes valuable insights with possible clinical implications for assisted reproductive technology (ART). The methodology is well structured, the statistical analyses appear appropriate, and the results are clearly presented. Nevertheless, several issues outlined below limit the clarity and strength of the study’s conclusions and warrant further consideration:

  1. The control group comprises patients with infertility due to male or tubal factors. While this approach is common in ART-based studies, differences in underlying infertility etiologies may introduce potential confounding effects. The authors should clarify whether and how this issue was accounted for in the analysis.
  2. The targeted metabolomics methodology is described in sufficient detail. However, additional information regarding quality control (e.g., use of technical replicates, internal standard validation, and coefficient of variation thresholds) would enhance reproducibility. Furthermore, it should be indicated whether adjustments for multiple testing (e.g., FDR correction) were applied, given that 20 amino acids were analyzed.
  3. The study reports reduced levels of amino acids such as serine and glycine in EM, with pathway analysis identifying glycine/serine metabolism as the most significantly affected. Recent studies (e.g., PMID: 39585647) suggest that this pathway is closely linked to oxidative stress, mitochondrial dysfunction, and immune regulation, processes that are central to aging biology. The authors are encouraged to expand their discussion to address how dysregulation of glycine/serine metabolism may mechanistically connect EM to these aging-related processes.
  4. The ROC analysis for asparagine (AUC = 0.76) is encouraging, but the discriminative power remains modest. The authors should comment on the clinical feasibility of using these metabolites as biomarkers, particularly in combination, rather than as individual markers.
  5. While the manuscript is generally well written, minor language polishing would improve overall readability and clarity.

Reviewer 4 Report

Comments and Suggestions for Authors

Very good article

Author Response

We would like to thank our reviewer for the favorable comments

Reviewer 5 Report

Comments and Suggestions for Authors

The MS entitled “Follicular Fluid Amino Acid …” authored by Kurdi et al characterized the alterations in the amino acid composition of the follicular fluid in endometriosis which may help to individualize in vitro fertilization approaches. This study is of interest; however, there are some issues needs to be improved before the publication.

General comment

The title highlights the oxidative stress, are there any evidence for the oxidative indexes derived from the EM and the control? Or the authors detected the oxidative indexes directly such as SOD, ROS, MDA which can be performed?

Specific comments:

Regarding the grouping, “L109 A total of 56…of whom 17 achieved pregnancy” means these 17 patients is EM group (L114), and the other 39 not pregnant without EM? It is not clear.

Table 1 & 2: mark the significant differences in two groups, such as Baseline estradiol, FSH dose during stimulation; in addition, units are needed for the hormone levels

Table 3: outcomes mean what? Pregnancy rate? If it is, Chi-square test is needed. In addition, the format of this table needs to change

L234, inset table 3 in this paragraph to show the comparison result. Same as below: age. Authors can put this information by notes under the table.

Figure title should be put under the corresponding figure.

Round 2

Reviewer 2 Report

Comments and Suggestions for Authors

The authors had made significant changes, and it is acceptable for publication.

Comments on the Quality of English Language

its fine.

Reviewer 3 Report

Comments and Suggestions for Authors

The authors have properly addressed my concerns in the revised manuscript. I have no further comments.